# Altered Levels of Desaturation and ω-6 Fatty Acids in Breast Cancer Patients’ Red Blood Cell Membranes

**DOI:** 10.3390/metabo10110469

**Published:** 2020-11-17

**Authors:** Javier Amézaga, Gurutze Ugartemendia, Aitziber Larraioz, Nerea Bretaña, Aizpea Iruretagoyena, Joana Camba, Ander Urruticoechea, Carla Ferreri, Itziar Tueros

**Affiliations:** 1AZTI, Food and Health, Parque Tecnológico de Bizkaia, Astondo Bidea, 609, 48160 Derio, Bizkaia, Spain; jamezaga@azti.es; 2Onkologikoa Foundation, Paseo Doctor Begiristain, 121, 20014 San Sebastián, Gipuzkoa, Spain; anderu@onkologikoa.org (G.U.); gugartemendia@onkologikoa.org (A.L.); alarraioz@onkologikoa.org (N.B.); nbretana@onkologikoa.org (A.I.); airureta@onkologikoa.org (J.C.); jcamba@onkologikoa.org (A.U.); 3ISOF, Consiglio Nazionale delle Ricerche, Via Piero Gobetti, 101, 40129 Bologna, Italy

**Keywords:** arachidonic acid, breast cancer, linoleic acid, membrane lipidome, omega 6, red blood cell, SCD1

## Abstract

Red blood cell (RBC) membrane can reflect fatty acid (FA) contribution from diet and biosynthesis. In cancer, membrane FAs are involved in tumorigenesis and invasiveness, and are indicated as biomarkers to monitor the disease evolution as well as potential targets for therapies and nutritional strategies. The present study provides RBC membrane FA profiles in recently diagnosed breast cancer patients before starting chemotherapy treatment. Patients and controls were recruited, and their dietary habits were collected. FA lipidomic analysis of mature erythrocyte membrane phospholipids in blood samples was performed. Data were adjusted to correct for the effects of diet, body mass index (BMI), and age, revealing that patients showed lower levels of saturated fatty acids (SFA) and higher levels of monounsaturated fatty acid, cis-vaccenic (25%) than controls, with consequent differences in desaturase enzymatic index (∆9 desaturase, –13.1%). In the case of polyunsaturated fatty acids (PUFA), patients had higher values of ω-6 FA (C18:2 (+11.1%); C20:4 (+7.4%)). RBC membrane lipidomic analysis in breast cancer revealed that ω-6 pathways are favored. These results suggest new potential targets for treatments and better nutritional guidelines.

## 1. Introduction

Cancer is currently the major cause of mortality and morbidity worldwide. In 2018, the incidence of the disease resulted in 18.1 million of new cases and 9.6 million of deaths [1]. Regrettably, the forecasts indicate a huge increase in these values, with 29.5 million new cases and 16.5 million deaths in 2040 [2]. Thus, efforts on finding new strategies and solutions are needed to face difficulties that the disease presents. Among them, the importance of the cell membrane emerges as an essential structure for cell replication that modulates its composition and participates in survival, proliferation and migration processes in cancer. Therefore, in recent years, membrane has been proposed as a promising target that may help current therapies [3,4,5].

Phospholipids are the major organizational components of membranes that respond to pathological and nutritional situations [6]. Dietary habits can play an important role by modifying membrane lipid composition through triglyceride or phospholipid intakes and the remodeling process known as Lands’s cycle [7]. In particular, the fatty acid residues of phospholipids are regulators of membrane physical properties such as permeability and fluidity, and some of them are effectors of signaling cascades such as inflammation, immunoregulation, and proliferation. Fatty acids can be saturated (0 unsaturation), monounsaturated (1 unsaturation), and polyunsaturated (>1 unsaturation), and only the first two types can be de novo synthesized. On the other hand, polyunsaturated fatty acids (PUFA), distinct in omega-3 (ω-3) and omega-6 (ω-6) fatty acids (FA), are essential for humans, since their precursors (linoleic and alpha-linolenic acids, respectively) must be obtained from the diet. Once the precursors are in the organism, they can be metabolized by enzymatic systems to form other PUFA that have important physiological functions as mediators of several processes, as previously mentioned [8,9]. In cancer, cells adapt their membrane properties to sustain their survival with a variability that depends on the stage and type of cancer [10]. To sustain cell replication, exogenous and endogenous lipids are needed, and lipid metabolism is altered by prioritizing de novo synthesis and forming saturated fatty acids (SFA) from carbohydrates [11]. On the other hand, cells must correct the rigidity of SFA to impart the optimal membrane fluidity by partial transformation into monounsaturated fatty acids (MUFA), thus reducing rigidity as a strategy of resistance to treatments with decreased permeability to drug intake [10,12]. Finally, polyunsaturated fatty acids (PUFA) play important roles by which they modulate different processes such as lipid peroxidation and cell oxidative stress [13,14], modulation of signal transduction pathways involved in inflammation [15,16], and regulation of gene expression for controlling growth factor-mediated carcinogenesis [17].

Although dietary intakes have been correlated with metabolic status [18,19], intervention and epidemiological studies considering dietary factors are challenging due to the complexity of eating patterns. In this context, FA evaluation in red blood cell (RBC) membranes represents an optimal tool since all FA families are represented and are good reporters about the general condition of the tissues that result from the interaction among dietary, metabolic, and genetic factors. RBC can be obtained by non-invasive methods and cheap procedures. Our recent review of the state-of-the-art on fatty acids in oncology [9] highlighted that the RBC fatty acids are emerging as important biomarkers. In our previous study, we found metabolic changes associated with de novo synthesis of FAs and promoted metabolism of ω-6 FAs in a patient cohort with different cancer types and different chemotherapy treatments. 

In the present exploratory study, we focused on breast cancer patients that had not yet received any chemotherapy cycle. Mature RBC membranes were isolated and fatty acid derivatization was carried out by a fully automatized procedure to examine membrane fatty acid profiles and detect statistically relevant differences between breast cancer patients and healthy controls, together with a thorough control of dietary habits to envisage the specific disease contribution to this profile. The aim of this study was to foster further interest in membrane fatty acid profile as a comprehensive biomarker associated with cancer disease, with possible application to personalized nutritional strategies and monitoring during therapeutic treatments of cancer patients.

## 2. Results

### 2.1. Population Characteristics and Dietary Intake

A total of 39 controls and 45 patients with cancer took part in this study. Table 1 shows both participants characteristics and their dietary intake. As controls were body mass index (BMI) matched, there were not any differences between groups. However, in the case of age, patients were on average 9 years older. Regarding the human epidermal growth factor receptor 2 (HER2) status, (64.4 %) of cancer patients were HER2− and 35.6% were HER2+.

Table 1 shows information about participants’ dietary habits, including nutrient and food group intake. Patients showeda higher daily intake of calories, with differences (*p* < 0.05) in the consumption of lean fish, olive and sunflower oil, and eggs, and lower consumption of red meat. However, no differences in intake of carbohydrates, proteins, fiber, alcohol, or lipids were reported. Regarding FAs, several SFA (C16:0 and C18:0); several MUFA (C16:1 and oleic acid (OA)); and total PUFA, including the ω-3 FAs eicosapentaenoic acid (EPA), docosapentaenoic acid (DPA) and docosahexaenoic acid (DHA), and ω-6 FAs linoleic acid (LA) and arachidonic acid (AA), were higher in cancer patients.

### 2.2. Red Blood Cell Membrane Fatty Acid Profile

FA levels were adjusted by taking in account BMI, age, HER2 status, and dietary intake. Before analysis of covariance (ANCOVA) adjustment, a principal component analysis (PCA) model was performed to simplify the information about dietary intake and ease the interpretation. We carried out a three-factor model that included 15 items (carbohydrates, proteins, lipids, fats, FAs indicated in Table 1, and trans FAs) with the Kaiser-Meyer-Olkin test (KMO) of 0.78 and 81.9% of explained variance, indicating a good sampling adequacy for the analysis [20]. For the ANCOVA test, we considered as covariables the BMI, age, HER2 status, the three factors from the PCA, and the rest of dietary components from Table 1 that were not introduced in PCA. Results from FA analysis and ANCOVA adjustment are presented in Table 2. There were several differences between unadjusted and adjusted models. Multiple FA and indexes (16:1; *9c*, OA, DHA, total MUFA, *n*-3 cardiovascular risk index, and Stearoyl-coenzyme A Desaturase 1 (SCD1) 16:0/16:1) that were statistically significant were not significant after applying the correction factor. On the other hand, some other values did not change after adjustment and maintained their differences. First, SFAs and total trans FAs were lower (SFA: −8.3%; total trans FAs: −48.6%), whereas PUFAs showed an increase (7.9%) in patients. Considering specific FAs, patients had lower values of C16:0 (−7.5%), C18:0 (−9.2%), trans FAs 18:1 (−63.2%), and trans FAs 20:4 (−44.4%), and higher values of 18:1; 11c (25.0%), LA (11.1%), and AA (7.4%). LA and AA resulted in greater amounts over the total ω-6 (9.0%). No significant changes were found for ω-3 FAs. Concerning the indexes and ratios, the SFA/MUFA was lower in cancer patients (−12.2%), and the unsaturation (6.5%) and peroxidation (6.5%) indexes were higher.

With respect to enzymatic activity, calculated by the ratio between FA substrate and metabolite, SCD1 showed a higher activity in the pathway of conversion from 18:0 to 18:1; 9c (13.1%), but not for conversion between 16:0 to 16:1; *9c*.

No significant differences were found in relation to different HER2 status.

## 3. Discussion

Cancer is a very complex disease due to the large number of factors involved. Cells develop a great capacity to survive and proliferate without any growth control and under stress conditions. They modify several mechanisms, such as metabolism of lipids, carbohydrates, proteins, and nucleotides for providing enough precursors to maintain the functionality of the structures and functions [21,22,23]. In the cell replication process, the need of new membranes is a priority, since, although it is trivial, no cell can exist without its membrane. This becomes especially important for cancer cells where the recruitment of fatty acids for membrane lipids cannot be based only on the biosynthesis, and in fact it has been shown that phospholipids with one fatty acid tail (lysophospholipids) are scavenged from serum lipids [24]. Therefore, studies to understand the characteristic properties and compositions of cell membranes in cancer are relevant to the understanding of the overall metabolic changes and also to new findings related to anticancer strategies. As an example, which is not exhaustive, it is well known that cancer cells activate the de novo synthesis of lipids, thus ensuring SFA for the formation of lipid rafts that are essential for protein dynamics in membranes and cell survival [25,26]. Due to its effective role as a reporter cell, we focused on analyzing mature RBC membranes to obtain a better picture of membrane lipid metabolism. Since erythrocytes exchange FAs with lipid membranes from other cell types, they can provide an excellent representation of the lipid compositions in tissues [27,28,29], where tumors may be found but cannot be easily accessed. 

We also took into careful consideration the food intake of each person and adjusted the contribution of FA from the diet on the observed fatty acid values found in the mature RBC. As in our previous study [30], we found that this cohort of cancer patients presented lower amounts of SFA (also lower palmitic and stearic acids as single SFA) compared to healthy controls, indicating that SFA can be recruited for a higher MUFA biosynthesis. Indeed, the SCD1 activity in the pathway of forming oleic acid from stearic acid was found to be significantly higher. The increase of MUFA biosynthesis in cancer was described in other studies analyzing RBC [31,32,33]. However, it is worth underlining that the SFA–MUFA pathway can be followed in the RBC membranes, either for the decrease of SFA or for the increase of MUFA. 

The study cohort differed in some points between groups. Although controls were matched, age could not be adjusted due to an indication from the ethics committee, which stated that all participants should be recruited from the Onkologikoa hospital. For this reason, the control group was largely made up of workers, making it difficult to ensure age similarity. On the other hand, food intake was also different. Patients with cancer tended to eat more fish, eggs, cold meat, and oil, whereas red meat intake was controlled. These differences can be seen also in their FA intake, but when we consider the proportion of each nutrient against the total calories, they did not show differences. For both cases, the mathematical adjustment of diet, BMI, and age minimized the influence of these factors and allowed us to interpret the FA levels in RBC membranes as a result of the metabolic derangement in cancer. Therefore, we could confirm the metabolic changes in patients with cancer due to the activation of SCD1 in the pathway of forming oleic acid from stearic acid, which decreased the SFA/MUFA ratio. This is a known strategy used by the cancer cells to ensure membrane fluidity in tissues [11,34,35]. The increase of one specific MUFA structure was evident for cis-vaccenic acid, as also previously noticed [30]. It is worth noting that cis-vaccenic acid has a very low presence in food, and thus it can be related exclusively to a change in an increased metabolism. The biological role of vaccenic acid in the context of cancer is not known yet, and the complete processes of MUFA metabolism is under study in cell cultures [36].

The second aspect is connected to ω-6 FAs. We found an altered metabolism of ω-6 FA with higher levels of LA and DGLA in patients with cancer compared to controls, indicating an increase of fatty acids that are currently considered to be therapeutic targets in breast cancer metastasis [37]. This result was highlighted also in our previous work on patients with cancer; however, due to the heterogeneous population with various types of cancer and different chemotherapy treatments, we could not draw conclusions at that stage. It is important to see that the present work on breast cancer patients who had not yet received any treatment cycle gave the indication that the ω-6 FA and the levels of LA and DGLA can be considered biomarkers associated with cancer status. LA is an essential fatty acid and is necessary to start the cascade of ω-6. AA, on the other hand, can be both introduced by the diet and synthesized endogenously from LA (Figure 1). AA is a major component of membranes that plays an important role in cancer development. After activation through phospholipase A2 and liberation in the cytoplasm, it produces different metabolites by action of cyclooxygenase (COX) enzymes. For example, prostaglandin H2 is an important mediator of cancer-associated inflammation, response to growth factors, tumor progression, and metastasis [9,37,38]. Thus, the higher level of AA RBC from patients compared to controls, which was not seen in a previous study [30], could be an indicator of an activity of lipid mediators that is necessary to cancer cells. The increased levels of LA and AA also influenced the significant increases of unsaturation and peroxidation indexes. In such indexes, it is important to remark that the contribution of omega-6 and omega-3 fatty acids is profoundly different in the resulting metabolic consequences. In our breast cancer patient cohort, we did not detect any differences compared to controls, especially taking into account that we performed adjustments in order to reduce confounding factors. Our results can be evaluated in light of a possible competition between ω-3 and ω-6 pathways to be used in the case of an increased presence of the latter fatty acid family in cancer patients. As a matter of fact, beneficial effects of ω-3 in contrast with ω-6 are described in cancer [7,39]; however, no supplementation with ω-3 FAs has thus far been performed after having evaluated the membrane profile of the patients. 

## 4. Materials and Methods 

### 4.1. Subjects and Study Design

A prospective observational study was carried out on a group of newly diagnosed breast cancer patients at the Oncology Outpatient Unit from Onkologikoa Foundation (San Sebastian, Spain). Potentially eligible patients were identified by nurses during outpatient consultation from July 2017 to November 2018. The research included women between 18 and 70 years old, a BMI less than 35, diagnosed with breast cancer, without any other diseases, and with a life expectancy of at least 1 year. The BMI was calculated with the World Health Organization Criteria. HER2 characteristics, age, and BMI were also collected by nurses. The healthy control group consisted of healthy women from Onkologikoa Hospital, matched for BMI and age. Those having suffered cancer in the last 5 years were excluded.

The study protocol was approved by the Gipuzkoa Clinical Research Ethics Committee (TUECAL2017) and accomplished according to the Declaration of Helsinki Good Clinical Practice guidelines. Written informed consent was obtained from all patients.

### 4.2. Nutritional Status and Dietary Intake

Dietary intake was measured before starting with the chemotherapy treatment by using a 137-item food frequency questionnaire (FFQ) validated for the Spanish population [40], with reference to the last 4–6 months. Two trained researchers completed the FFQ with patients within 20–25 min personal interviews. The nutrient composition of their intake was determined using DIAL software (UCM & Alce Ingeniería S.A., Madrid, Spain) (V 3.7.1.0) [41].

### 4.3. Red Blood Cell Membrane Fatty Acid Profile Analysis

The fatty acid composition of mature RBC membrane phospholipids was obtained before starting with their chemotherapy treatments, being taken from venous blood samples (approximately 2 mL) as described previously [30]. Briefly, they were collected in vacutainer tubes (Sarstedt, Germany) with ethylenediaminetetraacetic acid (EDTA) in the fasting state. After plasma separation by centrifugation (4000 rpm for 5 min at 4 °C), we used a fully automated facility present in the Lipidomic Laboratory of Lipinutragen srl to obtain the mature RBC fraction, isolate their membrane phospholipids, and transform them into the corresponding fatty acid methyl esters (FAMEs), as previously reported [42,43,44]. It is worth underlining that the high-throughput system isolates the cell fraction on the basis of high density of the aged cells [45] and the procedure is standardized in accordance to the cell diameter, which is reduced compared to the general population (Scepter 2.0, EMD Millipore, Darmstadt, Germany). After isolation of the cell fraction, the robotic platform performs other subsequent steps of cell lysis, isolation of membrane pellets, phospholipid extraction by the Bligh and Dyer method [46], and transesterification of FAMEs with a potassium hydroxide (KOH)/methyl alcohol (MeOH) solution (0.5 mol/L) for 10 min at room temperature with a final FAME extraction using *n*-hexane (2 mL). FAMEs were analyzed using capillary column gas chromatography (GC). GC analysis was run on the Agilent 6850 Network GC System, equipped with a fused silica capillary column Agilent DB23 (60 m × 0.25 mm × 0.25 μm) and a flame ionization detector, using published conditions. The FA cluster made of 10 fatty acids and 2 trans FA isomers was evaluated by calculating each FA as a percentage over the total FA cluster (relative %). GC peaks were identified as 97% of the total peaks present in the GC analysis by comparison with commercially available standards. 

### 4.4. Red Blood Cell Membrane Fatty Acid Cluster

The FA cluster was formed by 12 fatty acids, representative of the main building blocks of the RBC membrane glycerophospholipids and of the 3 FA families: SFAs, with palmitic acid (C16:0) and stearic acid (C18:0); MUFAs with palmitoleic acid (C16:1; *9c*), OA (C18:1; *9c*), and cis-vaccenic acid (C18:1; *11c*); *ω-3* PUFAs with EPA (C20:5) and DHA (C22:6) acids, and *ω-6* PUFAs with LA (C18:2), DGLA (C20:3), AA (C20:4), and trans isomers, considering elaidic acid (C18: 1; *9t*) and mono-trans arachidonic acid isomers (mono trans-C20:4; *ω-6*) [47].

Considering these fatty acids, different indexes were calculated [34]: inflammatory risk index (% omega 6/% omega 3), cardiovascular risk index (% EPA + % DHA) [48], SFA/MUFA (%SFA/% MUFA), PUFA balance ((% EPA + % DHA)/total PUFA × 100) [49], free radical stress index (C18:1; *9t* + mono trans-C20:4; *n-6*), unsaturation index (UI) ((% MUFA × 1) + (% LA × 2) + (% DGLA × 3) + (% AA × 4) + (% EPA × 5) + (% DHA × 6)), and peroxidation index (PI) ((% MUFA × 0.025) + (% LA × 1) + (% DGLA × 2) + (% AA × 4) + (% EPA × 6) + (% DHA × 8)). Additionally, the enzymatic indexes of elongase and desaturase enzymes, the two classes of enzymes of the MUFA and PUFA biosynthetic pathways, were estimated by calculating the product/precursor ratio of the involved FAs: *delta*-6-desaturase + elongase (∆6D + ELO 18:2/20:3), *delta*-5-desaturase (∆5D 20:4/20:3), and two pathways of SCD1 (18:0/18:1 and 16:0/16:1).

### 4.5. Statistical Analysis

Normal data distribution was verified using a Shapiro-Wilk test, and those values that were not normal were log-transformed to normalize them. Differences between groups for the population characteristics (BMI, age), nutrient and food intake, and unadjusted FA levels were tested with a Mann-Whitney *U* test for those data that were not normally distributed and a two-tailed *t*-test for normally distributed variables. In addition, the FA levels from both groups were compared with ANCOVA, adjusting for age, nutrient intake, and BMI. A PCA was conducted on nutrient intake variables to reduce and simplify the dimension of these variables. Generated components were rotated by an oblique rotation (direct oblimin) to increase interpretability. The KMO and Bartlett’s test of sphericity were used to verify the sampling adequacy for the analysis. With an eigenvalue cut-off > 1, component interpretability and screen plot were used to decide the number of factors to retain. These components were included in the ANCOVA analysis. The level of significance was set at *p* < 0.05. All statistical analyses were performed using SPSS (IBM Corp. V 24.0, Armonk, NY, USA).

## 5. Conclusions

On the basis of the effect that the membrane FA balance can have on cancer cells and considering the essentiality of some fatty acids, it is necessary to rely on tools that can link food intake with the metabolism of the disease. The results of this study revealed different profiles of RBC membranes in breast cancer patients compared to controls, independently from the influence from their dietary habits. In comparison with our previous study, where different cancer types were studied, the results obtained for a homogeneous population group such as breast cancer were similar, pointing out de novo FA synthesis and synthesis of ω-6 FA as the main altered pathways. In this context, with additional research, targets for novel therapeutic interventions focused on FAs could be discovered.

## Figures and Tables

**Figure 1 metabolites-10-00469-f001:**
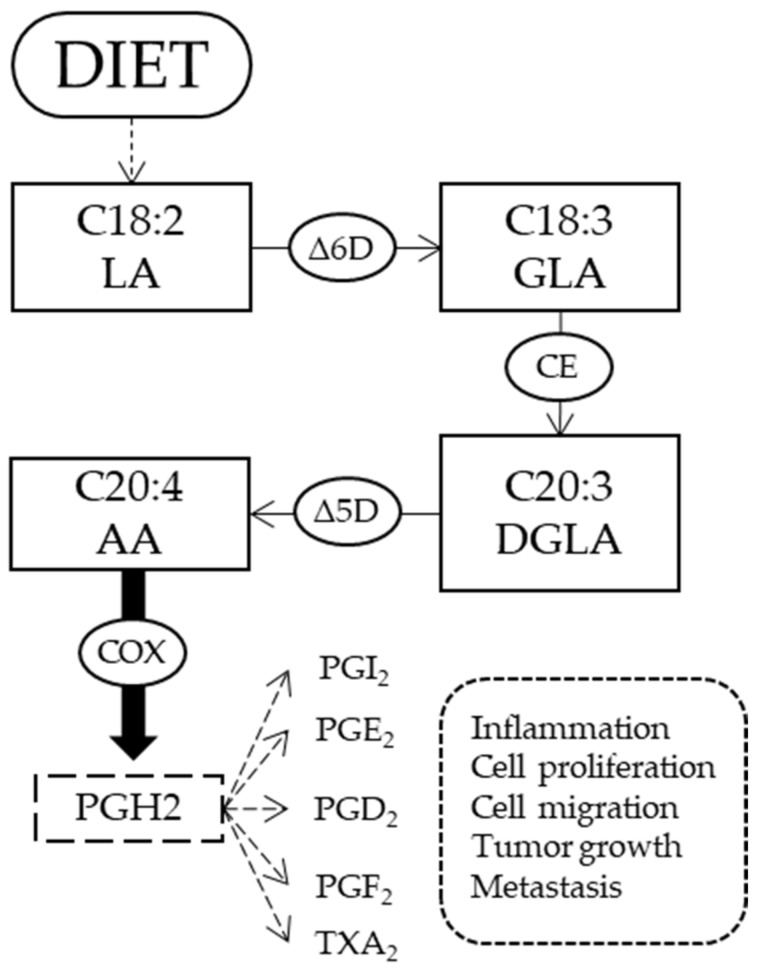
Metabolic pathway of ω-6 fatty acids. Linoleic acid (LA) comes from diet, and through ∆-6 desaturase enzyme forms gamma-linoleic acid (GLA). Then, an elongase (CE) adds two carbon atoms to form dihomo-γ-linolenic acid (DGLA) that, with the action of ∆-5 desaturase, produces arachidonic acid (AA). AA through cyclooxygenase (COX) enzymes is the precursor of some prostaglandins (PG), as the H_2_, I_2_, E_2_, D_2_, F_2_ and the thromboxane (TX) A_2_ involved in different processes associated with cancer.

**Table 1 metabolites-10-00469-t001:** Nutrients and food intake in cancer and control groups.

Characteristics/Intake	Total Nutrient/Food Intake		
	Control (*n* = 39) Mean ± SE	Cancer (*n* = 45) Mean ± SE	*p*-Value
Population characteristics			
HER2-		29 (64.4 %)	
HER2+		16 (35.6 %)	
Age (years)	43.08 ± 2.16	52.84 ± 1.40	<0.001
BMI (kg/m2)	23.93 ± 0.44	24.20 ± 0.41	0.650
Daily Energy and Nutrient Intake			
Calories (Kcal/day)	2068.97 ± 122.39	2539.24 ± 81.71	0.002
Carbohydrates (% of energy/day)	38.13 ± 1.03	36.09 ± 0.81	0.116
Simple sugars (g/day)	108.54 ± 8.24	101.68 ± 4.89	0.477
Protein (% of energy/day) *	17.54 ± 0.46	16.67 ± 0.14	0.221
Fiber (g/day) *	31.49 ± 2.34	34.70 ± 2.15	0.099
Alcohol (g/day) *	4.14 ± 0.93	11.25 ± 2.52	0.134
Fat (% of energy/day)	39.67 ± 0.96	41.42 ± 0.80	0.162
SFA (% of energy/day)	11.00 ± 0.35	10.71 ± 0.29	0.519
C14:0 (g/day)	2.00 ± 0.17	2.12 ± 0.17	0.661
C16:0 (g/day)	13.19 ± 0.84	17.51 ± 0.74	<0.001
C18:0 (g/day)	5.66 ± 0.40	7.31 ± 0.36	0.001
MUFA (% of energy/day)	18.60 ± 0.63	20.05 ± 0.55	0.085
C16:1 (g/day)	1.19 ± 0.08	1.59 ± 0.06	0.001
C18:1 (g/day) *	40.48 ± 2.86	52.77 ± 1.90	<0.001
PUFA (% of energy/day)	6.79 ± 0.33	6.81 ± 0.25	0.970
W3 (% of energy/day)	1.15 ± 0.06	1.16 ± 0.05	0.850
C18:3 (g/day)	2.04 ± 0.18	2.17 ± 0.11	0.154
C20:5 (g/day)	0.17 ± 0.02	0.32 ± 0.04	0.001
C22:5 (g/day) *	0.06 ± 0.01	0.09 ± 0.01	0.002
C22:6 (g/day)	0.36 ± 0.04	0.62 ± 0.05	<0.001
W6 (% of energy/day)	5.47 ± 0.28	5.45 ± 0.22	0.947
C18:2 (g/day)	12.71 ± 1.11	15.26 ± 0.84	0.015
C20:4 (g/day)	0.14 ± 0.01	0.18 ± 0.01	0.001
Food Groups (g/day)			
Oily fish	22.74 ± 2.84	27.05 ± 3.41	0.069
Lean fish	28.49 ± 3.01	50.06 ± 4.93	<0.001
Shellfish	16.82 ± 4.27	9.43 ± 1.29	0.069
Olive oil	31.72 ± 2.58	45.26 ± 2.18	<0.001
Sunflower oil	0.30 ± 0.11	2.36 ± 0.82	0.037
Nuts	14.42 ± 2.47	13.81 ± 2.02	0.522
Fruit	302.08 ± 33.83	310.85 ± 26.05	0.263
Vegetables	315.09 ± 30.35	306.27 ± 14.90	0.422
Dairy products	404.47 ± 35.11	358.14 ± 37.28	0.290
Eggs	20.04 ± 2.55	31.90 ± 1.69	0.002
Red meat	35.96 ± 4.44	30.33 ± 2.89	<0.001
Cold meat	15.61 ± 2.00	28.81 ± 2.41	<0.001

* Not normally distributed variables. SE: standard error.

**Table 2 metabolites-10-00469-t002:** Red blood cell (RBC) membrane fatty acid (FA) levels in cancer and control groups.

RBC Membrane FA (% rel)	Unadjusted Control	Unadjusted Cancer	*p*-Value	Adjusted Control	Adjusted Cancer	*p*-Value	Difference (%)
	(*n* = 39)	(*n* = 45)		(*n* = 39)	(*n* = 45)		
Saturated Fatty Acids							
Palmitic acid (16:0)	24.5 ± 0.4	22.6 ± 0.2	<0.001	24.5 ± 0.3	22.6 ± 0.3	<0.001	−7.5
Stearic acid (18:0)	19.8 ± 0.3	18.2 ± 0.2	<0.001	19.9 ± 0.3	18.1 ± 0.3	<0.001	−9.2
Monounsaturated Fatty Acids							
Palmitoleic acid (16:1; 9c) *	0.5 ± 0.0	0.4 ± 0.0	0.002	0.5 ± 0.0	0.5 ± 0.0	0.258	−10.0
OA (18:1; 9c) *	16.4 ± 0.3	17.4 ± 0.2	0.005	16.7 ± 0.3	17.1 ± 0.2	0.283	2.6
Cis-vaccenic acid (18:1; 11c *)	1.2 ± 0.0	1.4 ± 0.0	<0.001	1.2 ± 0.0	1.5 ± 0.0	<0.001	25.0
Polyunsaturated Acid							
LA (18:2 ω-6)	12.1 ± 0.2	12.8 ± 0.2	0.035	11.8 ± 0.3	13.1 ± 0.2	0.001	11.1
DGLA (20:3 ω-6) *	1.9 ± 0.1	2.0 ± 0.1	0.374	1.9 ± 0.1	2.0 ± 0.1	0.186	11.1
AA (20:4 ω-6) *	17.3 ± 0.3	18.2 ± 0.2	0.017	17.1 ± 0.3	18.4 ± 0.3	0.006	7.4
EPA (20:5 ω-3) *	0.8 ± 0.1	0.9 ± 0.1	0.076	0.8 ± 0.1	0.8 ± 0.1	0.971	0.0
DHA (22:6 ω-3)	5.4 ± 0.2	6.0 ± 0.2	0.042	5.6 ± 0.2	5.8 ± 0.2	0.588	3.2
Trans Fatty Acids							
Trans 18:1 *	0.2 ± 0.0	0.1 ± 0.0	<0.001	0.2 ± 0.0	0.1 ± 0.0	<0.001	−63.2
Trans 20:4 *	0.2 ± 0.0	0.1 ± 0.0	<0.001	0.2 ± 0.0	0.1 ± 0.0	0.002	−44.4
Total Fatty Acids							
Total SFA	44.4 ± 0.6	40.8 ± 0.2	0.009	44.4 ± 0.5	40.7 ± 0.4	<0.001	−8.3
Total MUFA	18.1 ± 0.2	19.2 ± 0.2	0.003	18.4 ± 0.3	19.0 ± 0.3	0.115	3.6
Total PUFA	37.4 ± 0.4	39.9 ± 0.3	<0.001	37.1 ± 0.4	40.1 ± 0.3	<0.001	7.9
Total omega 6	31.3 ± 0.4	33.0 ± 0.3	0.002	30.7 ± 0.4	33.5 ± 0.4	<0.001	9.0
Total trans *	0.4 ± 0.0	0.2 ± 0.0	<0.001	0.4 ± 0.0	0.2 ± 0.0	<0.001	−48.6
Fatty Acid Indexes—Ratios							
SFA/MUFA	2.5 ± 0.1	2.1 ± 0.0	0.002	2.5 ± 0.1	2.2 ± 0.1	0.001	−12.2
Unsaturation index	153.3 ± 1.8	164.1 ± 1.1	<0.001	153.7 ± 1.6	163.7 ± 1.4	<0.001	6.5
Peroxidation index	133.2 ± 2.0	143.5 ± 1.6	<0.001	134.3 ± 2.0	142.5 ± 1.8	0.007	6.5
Inflammatory risk index *	5.8± 0.5	5.1 ± 0.2	0.625	5.3 ± 0.4	5.5 ± 0.4	0.747	3.6
PUFA balance	16.5 ± 0.7	17.3 ± 0.6	0.337	17.4 ± 0.7	16.5 ± 0.6	0.390	−5.1
n-3 cardiovascular risk index	6.1 ± 0.3	6.9 ± 0.2	0.036	6.5 ± 0.3	6.6 ± 0.3	0.653	2.8
Enzymatic Indexes							
SCD1 18:0/18:1 *	1.2 ± 0.0	1.1 ± 0.0	<0.001	1.2 ± 0.0	1.1 ± 0.0	0.003	−13.1
Δ6D+ELO 20:3/18:2	0.1 ± 0.1	0.1 ± 0.1	0.904	0.1 ± 0.1	0.1 ± 0.1	0.893	1.0
Δ5D 20:4/20:3	9.8 ± 0.4	9.92 ± 0.4	0.796	9.9 ± 0.5	9.8 ± 0.5	0.889	−1.1
SCD1 16:0/16:1 *	0.0 ± 0.0	0.0 ± 0.0	0.037	0.0 ± 0.0	0.0 ± 0.0	0.524	7.3

Data are expressed with means ± standard error. Adjusted results are FA levels controlled by age, nutrient intake, and body mass index (BMI). * Not normally distributed variables. OA: oleic acid; LA: linoleic acid; DGLA: dihomo-γ-linoleic acid; AA: arachidonic acid; EPA: eicosapentaenoic acid; DHA: docosahexaenoic acid; SFA: saturated fatty acids; MUFA: monounsaturated fatty acids; PUFA: polyunsaturated fatty acids; SCD1: Stearoyl-coenzyme A desaturase 1; ELO: elongase.

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
