# Peer review of "Altered Levels of Desaturation and ω-6 Fatty Acids in Breast Cancer Patients’ Red Blood Cell Membranes"

_metabolites, 2020, doi:10.3390/metabo10110469_

Round 1
Reviewer 1 Report
Altered levels of desaturation and ω-6 fatty acids in breast cancer patient’s red blood cell membranes By Amezaga et al.
The study is about exploring the membrane fatty acid profile as a biomarker associated with breast cancer. This paper is well written and informative and brings attention to the value of studying the red blood cells as surrogates of disease status, nutrition, and therapeutic treatments
The study is straight forward and well analyzed; however, a few items need clarification before publication:
1- It is unclear how good the RBC reflect the lipid status of the tumor, is there any correlation?
2- Since Aromatase inhibitors (AI) are associated with joint pain associated with increased PUFAs, were there any associations in the study data with AI use? This could be very relevant as well with the use of lipids of RBC as biomarkers for breast cancer treatment.
3- Cholesterol is very abundant in RBC and changes with age (PMID 2014922) The authors do not mention any data related to cholesterol in this study, it would be very informative if these data were also available, and its relationship to saturation index.
4- Table 2: The ratio shown for SCD1 should be labeled as 18:0/ 18:1? Please clarify, the OA is increased in the cancer patients.
Reviewer 2 Report
The manuscript summarises the lipidomic analysis of patients’ RBCs and compares it with BMI matched healthy controls. The age of patient group is significantly different from control group, which the authors need to emphasize in more detail as why their choice of control group is different.
There are several language improvements needed , I could find the following examples (not limited to these, there may be many)
Page 2 line 88
gathers – shows?
Page 2 line 88
It is Organized??
Page 4 line 109-110
Collected in??
Page 5 line 133
What is meant by - Cancer disease is a very complex environment due to the large number of factors involved
Page 5 line 137
The need of new membranes is prioritized over all the other processes – any evidence for this statement - prioritization of membrane synthesis than any other cellular process - by authors (like reference)??
Page 7 line 243
What is meant by - The robotics performs also cell lysis isolation of 243 the membrane pellets
